# BMJ Open From embracing to managing risks

Justin Keen,[1] Emma Nicklin,[2] Nyantara Wickramasekera,[3] Andrew Long,[2] Rebecca Randell,[2] Claire Ginn,[4] Elizabeth McGinnis,[2] Sean Willis,[5] Jackie Whittle[5]

¹Leeds Institute of Health Sciences, University of Leeds, Leeds, UK
²University of Leeds, Leeds, UK
³University of Sheffield, Sheffield, UK
⁴NHS England, Leeds, UK
⁵Leeds Teaching Hospitals NHS Trust, Leeds, UK

**Correspondence to**
Dr Justin Keen;
J.Keen@leeds.ac.uk

## ABSTRACT

**Objective** To assess developments over time in the capture, curation and use of quality and safety information in managing hospital services.

**Setting** Four acute National Health Service hospitals in England.

**Participants** 111.5 hours of observation of hospital board and directorate meetings, and 72 hours of ward observations. 86 interviews with board level and middle managers and with ward managers and staff.

**Results** There were substantial improvements in the quantity and quality of data produced for boards and middle managers between 2013 and 2016, starting from a low base. All four hospitals deployed data warehouses, repositories where datasets from otherwise disparate departmental systems could be managed. Three of them deployed real-time ward management systems, which were used extensively by nurses and other staff.

**Conclusions** The findings, particularly relating to the deployment of real-time ward management systems, are a corrective to the many negative accounts of information technology implementations. The hospital information infrastructures were elements in a wider move, away from a reliance on individual professionals exercising judgements and towards team-based and data-driven approaches to the active management of risks. They were not, though, using their fine-grained data to develop ultrasafe working practices.

### Strengths and limitations of this study

► There have been very few studies that focus on the production of information and its use in managing hospital services.

► This was an in-depth comparative study of the production and use of information in four hospitals, employing observations, interviews and document analysis.

► The study design did not allow us to evaluate the effects of developments on patient outcomes.

► The study was only able to capture developments over a limited period: further studies would shed light on the development of information infrastructures over time.

## BACKGROUND

A series of reports published since the turn of the millenium has highlighted problems with the quality and safety of acute hospital services in many countries.[1–3] While there is evidence of improvements in focused initiatives, it is generally agreed that there is considerable scope to provide higher quality and safer services overall.[4,5] The problems have generated a range of proposed responses over the last 15 years. A recurring theme concerns the need for cultural change in hospitals, away from a 'blame culture' and towards one where staff have the confidence to report mistakes and are able to learn from them.[6] Our interest in this article is in another long-standing proposal, investments in information technology (IT) infrastructures, to facilitate the capture, analysis and use of data about the quality and safety of services.[7–9]

In any hospital, implementing the proposal involves substantial changes in working practices. Staff in wards and departments will capture data electronically rather than on paper. Hospitals need staff with the skills needed to design and deploy IT systems, to manage and interpret clinical data and to support clinical teams in data-driven improvement initiatives. In practice, this is a considerable challenge. Many IT investments, including the high profile HITECH programme in the USA and the National Health Service (NHS) National Programme for IT in England, have experienced problems with both implementation and routine use in wards and departments.[10,11] There is also evidence that hospitals can lack the capacity to analyse or learn from quality and safety data that are captured in wards and departments.[12] For every leading site, there are others that still face challenges.[13]

This evidence notwithstanding, hospitals in the NHS in England have continued to invest in IT systems, including real-time systems—where data are widely available as soon as they are captured—for managing wards. They have also sought to improve the volume and scope of data to support more effective governance of services, prompted by a series of policies and reports from 2008 onwards.[14,15] We studied the development of data and IT infrastructures at four acute NHS hospitals in England for the period 2013–2016. We

were particularly interested in whether and how they influenced the management of patients' risks. Vincent and Amalberti[16] describe three principal approaches. The first is based on a reliance on the judgements of individual health professionals, responding to risks as they arise. The second reflects a shift to team-based working, where teams proactively manage patients' risks. In the third approach, which resonates with some quality improvement methods, hospitals use data to analyse their working practices and 'design out' risks, thus creating ultrasafe environments.

The article addresses two questions. First, how do hospitals develop information infrastructures for capturing and using data about the quality and safety of services? Second, how do they use the resulting data to monitor and manage quality and safety? We conclude that acute hospitals are developing effective infrastructures, both for the real-time management of wards and for management oversight of quality and safety. This is part of a wider transition, away from a reliance on individual doctors and other professionals relying on their judgements and towards a model where clinical teams actively manage risks.

## METHODS

We used the Biography of Artefacts approach.[17] IT systems in organisations develop over many years, typically in piecemeal fashion. New functions are added periodically and linked to existing systems, so that infrastructures—amalgams of a number of systems and the working practices of the people who use them—develop over time. New systems are added incrementally; users can adapt to them over long periods and become deeply embedded in the day-to-day work of an organisation. If we want to understand why systems are used in the ways they are today, then we need to understand their histories. Furthermore, because these infrastructures develop in different ways in different parts of an organisation, it is necessary to study them over time and in more than one place—where changes of interest are likely to occur. Observations are used, in multisite longitudinal case studies, to build up a picture of the ways in which the infrastructure and the wider organisation adapt to one another over time.

Fieldwork was undertaken in four acute NHS hospitals, each given a pseudonym to promote anonymity: Solo, Duo, Trio and Quartet. The sites were identified and recruited via a telephone survey of 15 acute hospitals undertaken in the autumn of 2014. Sampling was both purposive and pragmatic. It was purposive in that we sought to recruit hospitals that had deployed real-time ward information systems or had formal plans to implement them. It was pragmatic because we could only select from sites that were included in the survey, all of which were within reasonable travelling distance of our research base and that were willing to participate.

**Table 1** Observations of meetings by site (hours)

| Fieldwork observations (hours) | Solo | Duo | Trio | Quartet | Total |
|---|---|---|---|---|---|
| Board-level quality meetings | 18.5 | 23 | 18 | 22.5 | **82** |
| Directorate meetings | 16.5 | 3 | 0 | 10 | **29.5** |
| Ward working practices | 13 | 22 | 17.5 | 19.5 | **72** |
| Total | 48 | 48 | 35.5 | 52 | **183.5** |

In line with established ethnographic methods, we attached particular weight to direct observation of participants' working practices.[18 19] Board level quality meetings and directorate meetings were observed at all four sites between May 2015 and July 2016 (see table 1). At all meetings, a team member took contemporaneous notes, which were written up as soon as practicable afterwards. We also undertook semistructured face-to-face interviews, between April 2015 and September 2016, to explore the views of senior and directorate managers, and members of informatics and information teams, about the development and use of information infrastructures in their hospitals, including developments in the 2 years before 2015 (see table 2).[20–22]

### Patient and public involvement

The study had a patient and public involvement panel, which provided advice on aspects of our fieldwork methods and commented on the findings and their interpretation. They were not involved in recruitment or the conduct of the study.

Hospital board meetings were held monthly: papers were analysed for every third month (April, July, October and January) during the period April 2013–October 2016 to establish trends in the volume and content of quality and safety data provided. Data on mortality, reported incidents and complaints, vital signs, pain management, nutritional status and the NHS Safety Thermometer were used as tracers.[21]

**Table 2** Interviews conducted by site/agency

| Interviews (numbers) | Solo | Duo | Trio | Quartet | Total |
|---|---|---|---|---|---|
| Senior managers | 4 | 5 | 6 | 5 | **20** |
| Directorate managers | 2 | 1 | 1 | 2 | **6** |
| Informatics staff | 2 | 5 | 10 | 6 | **23** |
| Ward staff | 8 | 10 | 10 | 9 | **37** |
| **Total** | | | | | **86** |

Observations were also undertaken on two wards in each hospital. Morning handover meetings, and the subsequent use of electronic whiteboards in the 30–60 min after the meetings, were observed regularly during the fieldwork period. (The electronic whiteboards were large screens, typically mounted on walls near a nurses' station and showing summary details for each patient.) Detailed contemporaneous notes of staff practices were taken, focusing particularly on the use of whiteboards, these being outward manifestations of the information infrastructures at ward level. We were also interested in the sources of, and use of, information more generally, including 'soft intelligence' discussed during handovers. In addition, observers occasionally asked staff to explain their actions 'on the spot' when it seemed to be important for the study, for example, why a handover meeting had spent so long on a particular topic. Semistructured interviews were undertaken with ward clinical managers and staff.

Five cross-site accounts—mini-biographies—were developed of the work of board quality committees, information and informatics teams, directorates (sometimes also referred to as clinical or business units) and ward teams (focusing on nursing staff but including junior doctors and consultants). The analytical strategy was ethnographic. Data from direct observations were used to develop initial timelines for each setting in each hospital. Open coding of interview transcripts was undertaken, and the coded material was used to develop the narrative accounts.[23] The accounts were then compared and contrasted with one another and integrated to provide overall narratives for each setting.

## RESULTS

The overall trend in developments within hospitals was towards integrated technology infrastructures or, rather, two parallel and loosely coupled infrastructures.[24] The first involved the deployment of real-time ward management systems. These were either developed as discrete systems and then progressively linked to other systems (Trio) or were components of electronic health records programmes (Solo and Duo). The second development focused on data warehouses. These were computer servers that held a range of datasets derived from 'live' systems, including the newly deployed real-time systems, along with patient administration systems, pathology and other departmental systems. The warehouses were continuously updated—every 15 min up to daily, depending on the systems involved—but were separate from the systems used 'live' in wards and departments. They were used to curate data, including validation of datasets, preparation of routine reports and the creation of off-one reports for quality and other committees.

### Real-time ward management systems

Quartet did not deploy systems in the period of the study. The other three hospitals deployed them successfully. In all three, they were designed collaboratively, principally by local informatics teams and ward nurses, with medical staff less directly involved. The design process was iterative—some interviewees described it as agile—with informatics teams producing versions for 'pilot' use, which ward staff fed back on, leading to design modifications until staff were happy with the systems.

Once deployed, junior doctors, nurses and healthcare assistants (HCAs) used tablets or laptops to capture data by the bedside. Some of the data were common across the three hospitals, notably nursing observations, which were used to calculate National Early Warning Scores (NEWS) scores automatically.[25] All three were able to programme alerts for future nursing tasks, such as the next set of observations or the next risk assessment. Junior doctors used devices to view clinical data, including the results of tests and scans.

Broadly, clinical staff were positive about tablets and laptops. For example, at Duo, a nurse and a junior doctor respectively told us:

> If we were to take a phone call, we can update on here any information immediately so it's straight on the whiteboards, the doctors can see straight away, all of the team can see, and if we're asked any questions we've got all the information available. (Duo, ward nurse)

> If I need to check something I'm not having to go down to the doctor's office [and] go through the doctor's notes, everything's on here so I know for example if they've been for a test. (Duo, junior doctor)

However, some problems were noted. One was that there were too few devices on some wards. Another concerned the difficulties experienced when the system crashed. The system did not go down often, or for very long, but there were problems when it did:

> If that screen goes down you can't see when your patient's obs [observations] are due, what they were before, or anything. (Trio, ward nurse)

Electronic whiteboards, located in or near ward stations, were used throughout the observation period to view ward-wide data 'at a glance' at Duo and Trio. Nurses, HCAs and doctors used them to check when patients' observations were next due and to check the locations of patients when they came onto the ward. A ward sister observed that:

> You've got this huge thing telling you … it's just easier to see, it's so much clearer…you can see people's blood pressure dropping … we're just more aware, I just think it's really good. (Trio, ward sister)

There were no substantive changes in use during the observation period. In contrast, clinicians at Solo told us that staff did not look at the whiteboard very often, because most of the data (eg, NEWS, risk of fall and nutrition) were duplicated on their laptops and on handover

sheets, which were used before the arrival of electronic whiteboards. Quartet used wall-mounted dry-wipe whiteboards throughout the study. Their use, principally for identifying key clinical risks for each patient, using magnet symbols, of did not change.

## Nursing handovers and other meetings

The real-time systems were designed and deployed in the broader contexts of information-intensive processes on wards. Working practices, notably in handovers and patient safety huddles, were stable over the course of the study: we did not find evidence that the technologies disrupted clinical work. Across the four hospitals, similar data were used in handovers and huddles throughout the period of observation. At Solo, for example, throughout the study, nurses starting their shift had a printed paper handover sheet, which included summary patient history details, dietary information, patient assessments (eg, falls risks), current medications and NEWS. Staff also discussed information that was not available on the handover sheets, such as jobs needing to be done (eg, changing dressings) or how a patient was feeling (eg, a patient's scores were fine but he had reported that he did not feel well).

## Development of routine data infrastructures

There had been substantive developments in infrastructures for handling routine data at all four hospitals, which they reported as having commenced in 2011 or 2012. Interviewees pointed out that hospitals had captured and submitted substantial volumes of data to national bodies since the 1980s; they repurposed some of these data for use in internal management reports. The changes were reflected in developments in the scale and scope of data reported board quality committees. In April 2013, three of the four board-level quality committees received reports presenting trends in a limited number of routine data items on 1–2 sides of paper. The report at Trio was longer, at over 30 pages, presenting trends in a larger number of indicators. By October 2016, all four hospitals presented detailed reports, presenting large numbers of indicators, typically on 60–100 pages, with many dozens of graphs, charts and tables. Reasons given for these changes included the desire to address long-standing problems with the credibility of data—by creating a 'single source of truth'—and the recognition by boards of the importance of monitoring quality and safety.

The data in reports were managed by hospital information teams. Several of our interviewees commented on the fact that many indicators were counts—numbers of incidents, numbers of deaths in hospital and so on. It was argued that this was, in large part, because national bodies had long focused on activity data and accordingly that was the data available to hospitals. At the same time, the numbers of 'narrative reports', which combined routine data about a particular topic with a text commentary, increased in the course of the study. For the monitoring of complaints, for example:

… before we didn't really measure how quickly the complaints were turned around … [now] we have the turnaround time reported, and the themes … have we responded in the right way? and if not why not? … So that's been a huge turnaround for us in terms of complaint reduction and how our teams are managing complaint responses. (Quartet quality committee member)

These developments were not costless. There were a number of comments about the time that information teams had to spend on verifying data and on producing reports. For example, considerable effort had to be devoted each month to the collection and collation of nationally mandated NHS Safety Thermometer data, even though much of the data were already recorded in patient notes and in Datix (a system used widely to record information about incidents). One respondent observed that:

… out of the twenty days in a month which a person works … eighteen of those days at the moment are about data verification … we've got to get that down to three or four days. (Duo informatics lead)

## Board committees: use and value of data

The data available in committee papers, both in detailed 'information packs' and in papers on specific topics (eg, trends in mortality and initiatives to reduce the incidence of pressure ulcers) were used extensively throughout the period of observation. Quality committees used data for performance management, for assurance, to identify organisational risks and to identify opportunities for service improvement. The value of the data was highlighted when non-executive directors used it to challenge executives. In Solo, for example, they questioned the value of receiving data on serious incidents that had happened many months earlier and challenged 'what we will do' statements, wanting to know how improvement would be achieved and measured.

Data in reports were not the only source of information to senior managers. Interviewees told us about additional strategies many introduced between 2014 and 2016, for gathering intelligence. These included the introduction of weekly meetings where staff could raise any issues or concerns with the chief nurse and medical director. Non-executives also went on regular ward 'walkabouts':

You triangulate what you are receiving [in board reports] with … what people actually say and talk about. (Quartet quality committee member)

Board quality committee members indicated that they believed that the governance of the quality and safety of services had improved over time. At Solo:

As a Trust now compared to where we were pre-Francis, we are in a much stronger position in terms of the quality and quantity of the information we get. And

you can always ask for more. (Solo quality committee member)

Similarly at Trio:

… because we've got access to that information, to be able to detect, for example, a deteriorating position in a ward … much more speedily … We can respond and put measures in place to recover that position. (Trio senior nurse manager)

## DISCUSSION

This study focused on the development of large-scale information infrastructures over time, casting light on both their implementation and use. We found that two distinct information structures were developing: one characterised by the use of real-time data and the other of retrospective data. The retrospective data were aggregated into management reports and used in routine review of quality and safety; this served to rationalise the curation and use of hitherto disparate datasets that were being generated across the hospitals. The difficulties that Quartet faced with real-time systems remind us that these developments are far from straightforward.

The principal strengths of the study derive from the extent of the fieldwork and the use of evidence from three distinct sources—observations of working practices, interviews and document analysis. The findings complement those found in sociological studies of the work of board members and of clinicians with responsibility for quality and safety more generally.[26 27] They typically have little to say about the information that clinicians and managers use, how it is produced or how they use it to inform their deliberations. The main weaknesses of the study are those usually associated with this study design; we could not evaluate patient outcomes, and the period of observation was limited, so that later developments could not be captured.

There were marked changes in the availability of data to board-level committees and to middle managers in the period 2013–2016. Our findings indicate that data on the quality and safety of services were used at all four hospitals. Boards that received little routine data in 2013, and thus had to rely on oral reports in meetings and on informal communications, were using data extensively to review performance in 2015 and 2016. This did not lead to the abandonment of less formal management strategies: indeed, these also increased during the period of the study, reflecting a stronger focus on quality and safety within the hospitals.[28]

The findings serve as a corrective to the many negative accounts of IT-based deployments in hospitals and suggest that hospitals are making two significant transitions.[10 11] First, NHS hospital managers have long had to rely on financial and activity data. Since 2013, managers have increasingly had access to retrospective reports on a range of quality and safety measures as well and used them to monitor performance. Three of the four hospitals also had extensive real-time data systems, providing effective day-to-day control of quality and safety. A number of interviewees stressed that they had historically encountered problems with the credibility of management data. The general acceptance of the accuracy of routine quality and safety data is therefore indicative of a sea change in attitudes and working practices.

Second, all four hospitals were effecting a transition in their approach to the management of clinical risks and hazards. Viewed in the context of Vincent and Amalberti's safety framework, the hospitals are moving away from embracing risks—where there is a reliance on the judgements and coping strategies of individual health professionals—towards a model where ward teams are actively managing risks.[16] We suggest that effective information systems are a prerequisite for that active management. Less positively, there was limited evidence that hospitals were taking steps towards Vincent and Amalberti's third approach, where hospitals use data to 'design out' risks, and creating ultra-safe environments.

The question arising from the last point is: why were not the hospitals using data to create ultrasafe environments? The findings hint at a possible explanation, namely that data collection is substantially determined by regulatory bodies, pursuing their purposes, and hospitals have limited resources to devote to data that they capture and use for their own purposes. If the latter are the key data for quality improvement, hospitals' efforts will be hampered by limited resources. Future research might therefore focus on the appropriate balance of effort devoted to capturing and curating management and clinical information and, in particular, on specifying the information that is needed to support quality improvement initiatives.

**Contributors** JK, AL, RR, EM, CG, SW and JW conceived the study and developed the protocol. JK, EN, NW and AL undertook fieldwork, and they and RR undertook analysis. All authors contributed to drafting the article.

**Funding** This study was funded by the NIHR Health Service and Delivery Research (HS&DR) programme, project 13/07/68.

**Disclaimer** The views and opinions expressed therein are those of the authors and do not necessarily reflect those of the HS&DR programme, National Institute for Health Research, National Health Service or the Department of Health.

**Competing interests** None declared.

**Patient consent** Not required.

**Ethics approval** Ethical approval was obtained from the University of Leeds Faculty of Medicine and Health ethics committee.

**Provenance and peer review** Not commissioned; externally peer reviewed.

**Data sharing statement** No additional data available.

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
