## [Reviewer comments · BMJ Open]

ARTICLE DETAILS

TITLE (PROVISIONAL)	From embracing to managing risks: a biography of information infrastructures in English hospitals
AUTHORS	Keen, Justin; Nicklin, Emma; Wickramasekera, Nyantara; Long, Andrew; Randell, Rebecca; Ginn, Claire; McGinnis, Elizabeth; Willis, Sean; Whittle, Jackie

VERSION 1 – REVIEW

REVIEWER	Robin Williams The University of Edinburgh, UK
REVIEW RETURNED	17-Apr-2018

GENERAL COMMENTS	The paper makes a valuable contribution to our understanding of the development and evolution of information infrastructures in UK hospitals. The paper charts two important developments that have received little attention in existing literature: the emergence through relatively lightweight development methods of local infrastructures for capturing and sharing real time data for ward management; the development of data repositories enabling the collation of retrospective information to monitor performance. These represent an important empirical contribution, warranting publication (and perhaps this might be more strongly emphasised). The paper sees these developments as occasioning a shift in approaches to managing patients risks, conceptualised, following Vincent and Amalberti's safety framework, as a move from "embracing risks" (relying on the judgements and coping strategies of individual health professionals) towards a model where ward teams collectively "actively manage risks". Though Vincent and Amalberti's framework is introduced on the final page of analysis (and hinted at in the title), it would be helpful to bring this into the introduction if it is to be given such salience in the analysis. I have a reservation about the phrasing of bullet 6 in the article summary, that: The study would have benefitted from tracking developments over a longer period. The point about the biography of artefacts perspective is surely that even longitudinal studies of the sort undertaken here, given current research funding models, still only have a relatively limited timescale in relation to the long term processes surround the development of infrastructures and working practices. This may require for example additional follow-up studies.
---

REVIEWER	Dr SJ Armstrong University of the Witwatersrand, South Africa
REVIEW RETURNED	09-May-2018

GENERAL COMMENTS	This study is of great interest to all responsible for, and interested in improving quality in health care. It provides for an important review of practices relating to IT systems which have, as discussed, tended to evolve in a somewhat disparate manner rather than being purpose designed. This article provides an opportunity to reflect on the impact of the use of these IT systems on health care globally. The methods were clearly well thought out and implemented but it is difficult for the reader to fully understand how they were implemented with the short descriptions given and the assumption is made, maybe quite reasonably, that the reader should be familiar with the methods or read up independently on the methods. I appreciate the word limit is often a problem when writing an article on such a complex study. As an international reviewer not familiar with the practices within the NHS I found some statements quite difficult to understand. One such term related to white boards. I know what they are but it was only when I read further in the article that I understood how they were being used in the context of this study. The results section attempts to report on what must have been a huge amount of data from 4 different sites. It was skillfully categorised under four themes but due to the four sites responding often differently from one another, tended to discuss site by site under each heading and it was difficult to get an overall sense of practices in some aspects. The quotes added value and understanding to the results section. The discussion section is well balanced pointing out the positive and negative findings but I found myself wanting a conclusion as the article ends quite abruptly. Thank you for the opportunity to review such an interesting article.
--

VERSION 1 – AUTHOR RESPONSE

Reviewer 1

We are grateful for the positive comments about the paper.

Vincent and Amalberti not mentioned before the last page of the text - accepted, new text added to the Background section.

Phrasing of bullet point (actually number 5, rather than 6) - accepted, this has been edited to address the reviewer's comment.

Reviewer 2

Again, we are grateful for the positive comments.

Need to define key terms - 'whiteboards' are defined. We have checked the whole text, and believe that this is the only term that is not defined when it first appears.

Article ends abruptly - accepted. We have re-drafted the last two paragraphs, so that - we hope - the article arrives at a natural conclusion.

VERSION 2 – REVIEW

REVIEWER	Robin Williams The University of Edinburgh, UK
REVIEW RETURNED	09-Aug-2018

GENERAL COMMENTS	The authors have responded effectively to the various issues previously raised. I have no further critical comments. I hope that this valuable paper will appear without delay.
---

REVIEWER	SJ ARMSTRONG University of the Witwatersrand, South Africa
REVIEW RETURNED	09-Aug-2018

GENERAL COMMENTS	There is a typo graphical error on page 12 line 13. Hospital should be plural. I remain puzzled as to what "the prescription" line 22, page 6 refers to but this may be understood by others. This article will assist in raising awareness of the importance of studying the impact of the many initiatives we tend to introduce without due thought for the implications and provides evidence that the ones reported on, at least, do have some value.
--

VERSION 2 – AUTHOR RESPONSE

Thank you for your decision letter of 27th September, and further email of 8th October. We attach a marked and an unmarked copy of the paper, with the requested changes. We have -

- Edited the 'strengths and weaknesses' bullet points, and added text on strengths and weaknesses in the Discussion
- Added a final paragraph. We have also made minor editing changes in the Discussion, so that the correspondence to your list - main findings, strengths and weaknesses, meaning, etc - is clearer.
- We have proof read the attached version of the paper.
- We have made minor edits to address the points made by Reviewer 2.

We have also edited the Abstract, as requested in your email of 8th October.